# Functional Characterization of JcSWEET12 and JcSWEET17a from Physic Nut

**DOI:** 10.3390/ijms25158183

**Published:** 2024-07-26

**Authors:** Pingzhi Wu, Youting Wu, Zhu Yu, Huawu Jiang, Guojiang Wu, Yaping Chen

**Affiliations:** 1Key Laboratory of South Subtropical Fruit Biology and Genetic Resource Utilization, Ministry of Agriculture and Rural Affairs & Guangdong Provincial Key Laboratory of Science and Technology Research on Fruit Trees, Institute of Fruit Tree Research, Guangdong Academy of Agricultural Sciences, Guangzhou 510640, China; wupingzhi@gdaas.cn (P.W.); 15089877025@163.com (Z.Y.); 2Key Laboratory of South China Agricultural Plant Molecular Analysis and Genetic Improvement & Guangdong Provincial Key Laboratory of Applied Botany, South China Botanical Garden, Chinese Academy of Sciences, Guangzhou 510650, China; wuyouting0529@163.com (Y.W.); hwjiang@scbg.ac.cn (H.J.); wugj@scbg.ac.cn (G.W.)

**Keywords:** SWEET, flowering time, salt tolerance, physic nut

## Abstract

Physic nut (*Jatropha curcas* L.) has attracted extensive attention because of its fast growth, easy reproduction, tolerance to barren conditions, and high oil content of seeds. SWEET (Sugar Will Eventually be Exported Transporter) family genes contribute to regulating the distribution of carbohydrates in plants and have great potential in improving yield and stress tolerance. In this study, we performed a functional analysis of the homology of these genes from physic nut, *JcSWEET12* and *JcSWEET17a*. Subcellular localization indicated that the JcSWEET12 protein is localized on the plasma membrane and the JcSWEET17a protein on the vacuolar membrane. The overexpression of *JcSWEET12* (OE12) and *JcSWEET17a* (OE17a) in Arabidopsis leads to late and early flowering, respectively, compared to the wild-type plants. The transgenic OE12 seedlings, but not OE17a, exhibit increased salt tolerance. In addition, OE12 plants attain greater plant height and greater shoot dry weight than the wild-type plants at maturity. Together, our results indicate that JcSWEET12 and JcSWEET17a play different roles in the regulation of flowering time and salt stress response, providing a novel genetic resource for future improvement in physic nut and other plants.

## 1. Introduction

As a kind of renewable alternative energy, biodiesel has attracted much attention because of its good environmental effect and great economic potential. The biomass-enriched energy plant physic nut (*Jatropha curcas* L.), belonging to the Euphorbiaceae family, is a kind of plant suitable for the production of biodiesel and is one of the most suitable energy plants for biodiesel feedstock [1,2]. It often appears as a shrub or small perennial tree [2]. In addition, physic nut is a kind of multi-purpose, high-light-efficiency tree species, and suitable for planting in poor soil in marginal lands; these excellent characteristics make it widely concerned by people [3].

Sugars act not only as energy reserves to provide energy for plant activities but can also function as osmotic compounds to regulate the process of plant resistance [4,5]. Flowering is positively correlated with the accumulation of sugar in plant leaves, such as tomato (*Solanum lycopersicum*), potato (*S. tuberosum*), and Arabidopsis (*Arabidopsis thaliana*) [6,7,8]. To ensure normal growth and plant development, sugars move long distances from leaves to various sink organs using sugar transporters [9,10]. The import and export of hexose and sucrose is controlled by the passive transporter, SWEETs, along with concentration gradients of sugar [11,12,13]. Based on their evolutionary relationships, SWEETs could be classified into clades I–IV. Members of clades I or II, III, and IV preferentially transport hexoses, sucrose, and glucose or fructose, respectively [12,14]. 

SWEETs play roles in plant development stages, including phloem loading, seed filling, pollen development, and physiological processes of stress response [15,16]. Arabidopsis clade III members participate in freezing and cold tolerance [17], salt response [18], nectar secretion [19], flowering time [20], anther dehiscence, and seed development [21]. In rice, both SWEET11 and 15 contribute to seed filling, and SWEET13 and 15 participate in the response to abiotic stress mediated by ABA signaling [22,23,24]. Silencing of tomato *SlSWEET14*, a plasma membrane-localized protein that transports fructose, glucose, and sucrose, promotes plant height and fruit size [25]. In addition, both SlSWEET15 in tomato and GmSWEET15 in soybean regulate seed development [26,27]. The overexpression of *AtSWEET16*, a clade IV SWEET, can increase cold tolerance [28,29]. 

Many studies have shown that the SWEET gene regulates the distribution of carbohydrates in seeds and stress response and has a relatively significant potential value in creating varieties with enhanced storage or stress resistance. In a previous study, we identified a total of 18 *JcSWEET* genes in the physic nut genome and performed transcription analysis of these genes in different tissues and abiotic stresses [30]. In the current study, we selected a SWEET12 gene that is relatively highly expressed in seeds and a SWEET17a gene that responds to drought/salt stress to carry out research in order to obtain excellent gene resources. We examined the function of *JcSWEET12* and *JcSWEET17a* genes through their overexpression in Arabidopsis. The *JcSWEET12* gene is a member of clade III, while *JcSWEET17a* is a member of clade IV [28]. We cloned *JcSWEET12* and *JcSWEET17a* genes and studied the subcellular localization of the encoded proteins. After expression in Arabidopsis, the roles of *JcSWEET12* and *JcSWEET17a* in plant growth and development and saline/drought tolerance were analyzed. These results help in further characterizing the function of SWEET proteins, and provide a novel genetic resource for future improvements in physic nut and other plants.

## 2. Results

### 2.1. Subcellular Localization of JcSWEET12 and JcSWEET17a Proteins

To determine the action location of JcSWEET12 and JcSWEET17a proteins in cells, we analyzed their subcellular localization. The *JcSWEET12:GFP* and *JcSWEET17a:GFP* constructs were generated and agroinfiltrated into the *Nitotiana benthamiana* leaves. As a result, the fluorescence of the positive control GFP was found to be located in the cytoplasm and nucleus (Figure 1A). The green fluorescence of JcSWEET12:GFP was clearly observed on the plasma membrane, and it was located on the outer side of chloroplasts, which were distributed along the periphery (Figure 1B). To confirm these results, the *JcSWEET12:GFP* construct was co-introduced into the epidermal cells of *N. benthamiana* leaves with a plasma membrane marker (*AtPIP2A:mCherry*). Confocal images showed that GFP fluorescence signals overlapped with red fluorescence (Figure 1C). These results indicate that the JcSWEET12 protein was localized on the plasma membrane. In contrast, the JcSWEET17a protein was localized on the vacuolar membrane, indicated by the fact that the green fluorescence of JcSWEET17a:GFP did not overlap with the red fluorescence of AtPIP2A:mCherry and the red fluorescence layer could be clearly observed to be located between two layers of green fluorescence in several positions (Figure 1D). To further confirm the localization of the JcSWEET17a protein, a *JcSWEET17a:GFP* construct was co-transformed into Arabidopsis mesophyll protoplasts with a tonoplast membrane marker (*AtTPK:mCherry*). Confocal images showed that GFP fluorescence signals of JcSWEET17a:GFP overlapped with red fluorescence derived from AtTPK:mCherry. This result also indicates that the JcSWEET17a protein was localized on the vacuolar membrane (Figure 1E,F). 

### 2.2. Overexpression of JcSWEET12 Causes Later Flowering in Arabidopsis

To investigate the functions of the *JcSWEET12* gene, its coding sequences were overexpressed in *A. thaliana* under the control of a CaMV 35S promoter (OE12). Three independent T_3_ homozygous OE12 transgenic lines (OE12-5, OE12-12, and OE12-24) were used in the following study, and the expression levels in these lines were examined by semi-quantitative RT-PCR (Figure 2A). 

Under normal growth conditions, OE12 lines showed no significant differences from WT before bolting in terms of plant size (Appendix A), shoot dry weight (Appendix A), and leaf shape (Appendix A). However, the OE12 lines displayed a later flowering phenotype (Figure 2B). After 36 days of growth, all WT plants had bolted, while only 58.34% and 88.89% of plants of the two OE12 lines, OE12-12 and OE12-24, with higher expression levels had bolted (Figure 2C). 

### 2.3. Overexpression of JcSWEET12 Increases Shoot Biomass in Mature Stage Arabidopsis

Plants of the OE12 lines displayed similar sizes and shoot biomass to the WT before bolting, but OE12-12 and OE12-24 looked larger than the WT plants at the mature stage (Figure 3A). Compared to the wild-type plants, the OE12-12 plants were 12.4% taller, while the dry weight of shoots was 20.6% and 15.3% greater (Figure 3B,C). There were no significant differences in seed yield-related traits between OE12 lines and wild-type plants, i.e., inflorescence number (Appendix A), silique number (Appendix A), silique length (Appendix A), seed number per silique (Appendix A), and 1000-seed weight (Appendix A).

### 2.4. Overexpression of JcSWEET12 Increases Salt Tolerance in Arabidopsis

The wild-type (WT) and OE12 lines did not differ in their seed germination efficiency in relation to the sugars tested—sucrose, glucose, and fructose (Appendix A). To gain an insight into the function of *JcSWEET12* in response to salt/drought stress, the three OE12 lines and wild-type Arabidopsis were examined. No differences in plant size and shape were detected between WT and OE12 lines on a normal 1/2 MS medium (Figure 4A), whereas the survival rate of the OE12 lines was significantly higher than those of WT seedlings when cultivated with 150 mM NaCl (Figure 4A–C). In addition, the relative electrical conductivity of OE12-5 and OE12-12 lines was significantly lower than that of WT seedlings after salt stress treatment with 150 mM NaCl (Figure 4D,E). However, there were no significant differences in shoot size, color, or root length between WT and OE12 lines when cultivated with 300 mM mannitol (Appendix A). These results suggest that the overexpression of *JcSWEET12* in Arabidopsis could improve salt tolerance. 

### 2.5. Overexpression of JcSWEET17a Causes Early Flowering in Arabidopsis

To investigate the functions of the *JcSWEET17a* gene, its coding domain sequences were overexpressed in Arabidopsis under the control of a CaMV 35S promoter (OE17a). Three independent T_3_ homozygous OE17 transgenic lines (OE17a-8, OE17a-9, and OE17a-11) were used in the following study. The expression levels of the *JcSWEET17a* gene in OE17a-11 leaves were relatively higher than in OE17a-8 and OE17a-9 when examined by semi-quantitative RT-PCR (Figure 5A). Under normal growth conditions, there were no significant differences between WT and OE17a lines with respect to either plant size (Appendix A) or shoot dry weight (Appendix A) before bolting or to leaf shape at the bolting stage (Appendix A). However, the OE17a lines displayed an early flowering phenotype (Figure 5B). The OE17a plants bolted 2-3 days earlier than the WT plants (Figure 5C). Like the OE12 lines, the seed germination of OE17a lines shared the same response as the WT lines to sucrose, glucose, and fructose (Appendix A). The overexpression of *JcSWEET17a* in Arabidopsis did not improve salt tolerance (Appendix A).

### 2.6. Overexpression of JcSWEET17a Influences Silique Characters in Arabidopsis

The OE17a lines did not exhibit significant differences in plant height (Appendix A), shoot dry weight (Appendix A), main inflorescence number (Appendix A), and 1000-seed weight (Appendix A) compared to the wild-type plants at the mature stage. However, plants of OE17a-9 and OE17a-11 lines had more siliques per plant (Figure 6A). In addition, the average length of silique from OE17a-9 and OE17a-11 lines was longer than that of the wild-type by 2.4% and 5.5%, respectively (Figure 6B,C). The seed number per silique in OE17a-11 plants was 5.5% greater than that of the wild-type (Figure 6D). 

## 3. Discussion

The divergent biological functions of some SWEET genes have been characterized in a number of species. In the present study, we confirmed that the clade III member, *JcSWEET12*, and the clade IV member, *JcSWEET17a*, fulfil different functions in regulating flowering time and salt response. First, JcSWEET12 and JcSWEET17a function in different subcellular positions: the plasma membrane and vacuole membrane, respectively (Figure 1). Second, JcSWEET12 and JcSWEET17a have opposite roles in regulating flowering time when expressed in Arabidopsis (Figure 2 and Figure 5). Finally, JcSWEET12, but not JcSWEET17a, enhances salt tolerance (Figure 4). 

The clade III members are mainly located in the plasma membrane; the JcSWEET12 gene belongs to this clade (Figure 1), and subcellular localization analysis showed that JcSWEET12 is a plasma membrane sugar transporter, which is consistent with the subcellular localization of Arabidopsis AtSWEET12 [14]. The clade IV members are all vacuolar membrane-localized; however, some members with dual subcellular localization, like JcSWEET16 and SlSWEET15, could also be expressed in the plasma membrane [27]. Like AtSWEET16 and AtSWEET17 [28,31], JcSWEET17a is temporarily only found in the vacuole membrane.

Although also belonging to clade IV, JcSWEET16 may be associated with early flowering and salt tolerance in Arabidopsis plants [30]. This result suggests that JcSWEET16 and JcSWEET17a exhibit functional divergence in relation to abiotic stress tolerance. The overexpression of *CsSWEET17* (a clade IV member from *Camellia sinensis*) has been found to increase the leaf and seed sizes of the transgenic Arabidopsis [32], whereas the leaf size (Appendix A) and 1000-seed weight (Appendix A) were not significantly increased in OE17a Arabidopsis lines. It was found that AtSWEET17 was induced to express during the lateral root growth stage and under drought treatment. Under drought treatment, the number of lateral roots of an *atsweet17* mutant decreased, and the expression of transcription factors related to lateral root development was down-regulated, which resulted in reduced drought tolerance of *atsweet17* mutant plants [33]. In this study, there was no significant difference in root length of OE17a Arabidopsis seedlings under 300 mM mannitol treatment or normal conditions compared with the wild-type (Appendix A). These results suggest that functions of SWEET17 genes may have differentiated during the evolution of the species.

On the other hand, transgenic Arabidopsis lines overexpressing *DsSWEET12* (a clade III member from *Dianthus spiculifolius*) [34] but not *JcSWEET12* (Appendix A) were associated with a higher tolerance to osmotic stresses. *AtSWEET10*, a clade III member, was activated by the FLOWERING LOCUS T (FT) signaling pathway. The overexpression of *AtSWEET10*, but not *AtAWEET13*, accelerates flowering in Arabidopsis [20]. Transgenic Arabidopsis plants with *AtSWEET15* (*SAG29*, a clade III member) overexpression were hypersensitive to salt stress [18]. These different biological effects of SWEET overexpression in Arabidopsis indicate that SWEET members in the same clade or from different species exhibit functional divergence in their protein characteristics, such as function regulation at the post-translational level [12,35] and transported substances other than sugars. For examples of clade III SWEET proteins, AtSWEET13 and AtSWEET14 are able to mediate cellular GA uptake, which may be involved in modulating GA response in Arabidopsis [21]. Barley (*Hordeum vulgare*) HvSWEET11b could mediate the transfer of not only sucrose and glucose but also cytokinin [36]. 

The overexpression of *JcSWEET16* [30] and *JcSWEET17a* (Figure 5B,C), two membranes located in clade IV SWEET from physic nut, in Arabidopsis all caused early flowering, but the overexpression of other clade IV SWEET numbers, such as *AtSWEET16*, in Arabidopsis was not reported to affect flowering time. The EARLY RESPONSE TO DEHYDRATION6-LIKE4 (ERDL4) protein, a member of the monosaccharide transporter family, resides in the vacuolar membrane in Arabidopsis. *ERDL4*-overexpressing plants show larger rosettes and roots, a delayed flowering time, and increased total seed yield [37]. These results indicate that different kinds of sugar transporters in the vacuolar membrane exhibit diverse roles in regulating flowering in Arabidopsis. Taken together, our findings support a model for two SWEET genes regulating flowering time and abiotic stress tolerance (Figure 7). Further research is needed on the study of the functional mechanism of SWEET genes in regulating flowering time.

## 4. Materials and Methods

### 4.1. Plant Material and Growth Conditions

The Arabidopsis ecotype Columbia-0 (Col-0) was used as the wild-type (WT) in this study. After surface disinfection, the seeds were incubated for 2 days in the dark at 4 °C and were grown in a growth chamber under a long-day photoperiod (16 h light/8 h dark) at 22 ± 2 °C. 

### 4.2. RNA Isolation and Gene Cloning

Total RNA was extracted from seeds and leaves of physic nut using the CTAB method, and first-strand cDNA was synthesized as previously described [38]. The cDNA was used as a template for amplifying the *SWEET* genes with the specific primers listed in Appendix A. The PCR products were cloned into the pMD18-T vector (TaKaRa) and then sequenced. The isolation of total RNA from Arabidopsis leaves was carried out using Trizol reagent (Invitrogen, Burlington, ON, Canada) following the manufacturer’s instructions.

### 4.3. Expressing Plasmid Constructs

The *pMD18-T-JcSWEET* clones were used as templates for amplifying the coding regions of the *JcSWEET12* and *JcSWEET17a* genes to construct binary vectors with the specific primers listed in Appendix A. 

For protein subcellular localization analysis, the fragment of *JcSWEET* cDNA was amplified and constructed into the modified plant expression vector named pCAMBIA1301-35S-GFP (CaMV 35S promoter, GFP, and rbcs terminator were inserted into MCS of pCAMBIA1301 by *Eco*R I/Sac I, *Sal* I/*Pst* I, and *Pst* I/*Hind* III sites, respectively) through *Kpn* I/*Sal* I sites, and the constructs were named *pCAMBIA1301-35S:JcSWEET-GFP*.

For Arabidopsis transformation, cDNA was amplified and cloned into *Kpn* I/*Sal* I sites of a modified plant expression vector named *pCAMBIA1301-35S* (CaMV 35S promoter and rbcs terminator were inserted into MCS of pCAMBIA1301 by *Eco*R I/*Sac* I and *Pst* I/*Hind* III sites, respectively), and the constructs were named *pCAMBIA1301-35S:JcSWEET*.

### 4.4. Protein Subcellular Localization Analysis

For transient expression of the infusion proteins in tobacco (*N. benthamiana*) leaves, the resultant plasmid *pCAMBIA1301-35S:JcSWEET-GFP* was transformed into *Agrobacterium tumefaciens* strain GV3101. Tobacco transformation followed the method described by Sosso et al. [39]. To mark the specific position on the plasma membrane, *AtPIP2A:mCherry* was used for co-localization analysis [40]. To further confirm the localization of the JcSWEET17a protein, the *JcSWEET17a:GFP* construct was co-transformed into Arabidopsis mesophyll protoplasts with a tonoplast membrane marker (*AtTPK:mCherry*). Fluorescence was observed on a Leica TCS SP8 confocal laser scanning microscope [25]. 

### 4.5. Arabidopsis Transformation

For Arabidopsis transformation, the expression vector *pCAMBIA1301-35S:JcSWEET* was transformed into GV3101 and was then used to infect Arabidopsis, as described previously [41]. The single-insertion homozygous lines were chosen for the subsequent analysis. Expression levels of the transgenic lines were determined by semi-quantitative RT-PCR with the specific primers listed in Appendix A. Flowering time was scored by observing the bolting ratio (a bolting height of 0.5 cm was taken to indicate bolting). 

### 4.6. Salinity and Osmotic Stress Treatment of Transgenic Arabidopsis

For salinity treatment, transgenic and wild-type seeds were sown on one half-strength Murashige and Skoog (MS) medium (pH 5.7) containing 1.0% (*w*/*v*) sucrose and 1.0% agar (*w*/*v*) after surface disinfection. The plates were incubated in the dark at 4 °C for 2 days and then grown in a growth chamber. After growth for 4 days, similarly sized seedlings were transferred to new vertical or horizontal 1/2 MS medium plates supplemented with 100 mM and/or 150 mM NaCl, as described previously [30,42]. Surviving seedlings were counted, and photos were taken after 6 days of treatment [28]. For osmotic stress treatment, the 4-day-old seedlings were transferred to new vertical 1/2 MS medium plates supplemented with 0 mM or 300 mM mannitol. For the determination of electrolyte leakage (EL), shoots (ca. 100 mg fresh weight) of 14-day-old seedlings after 5 days of saline treatment were collected and incubated in 20 mL distilled water and incubated on a shaker at 30 °C for 6 h. Initial conductivity (EC1) was measured with a conductivity meter (DDS-11A, Jintan Yitong Electronics Co., Ltd., Changzhou, China). The tubes were further incubated in boiling water for 30 min, and the complete ion leakage (EC2) was measured after cooling to room temperature. The %EL was calculated as (EC1/EC2) × 100%. 

### 4.7. Statistical Analysis

All experiments included three biological replicates, and the data are presented as the means ± SDs (standard deviations). Significance tests were carried out using SPSS software (version 17.0) based on Student’s t-tests at *p* < 0.01 or *p* < 0.05. In some cases, significant differences were identified according to one-way ANOVA with post hoc Duncan testing; different letters above bars denote *p* < 0.05.

## 5. Conclusions

The present study reveals that *JcSWEET12* and *JcSWEET17a*, clade III and IV members of the SWEET family, respectively, have a different subcellular localization and play opposite roles in mediating flowering time. In addition, overexpressing *JcSWEET12* improves the salt resistance and increases the biomass of shoots at the late growth stage of the transgenic plants, which may help crop plants to develop improved crop phenotypes with enhanced productivity and resilience.

## Figures and Tables

**Figure 1 ijms-25-08183-f001:**
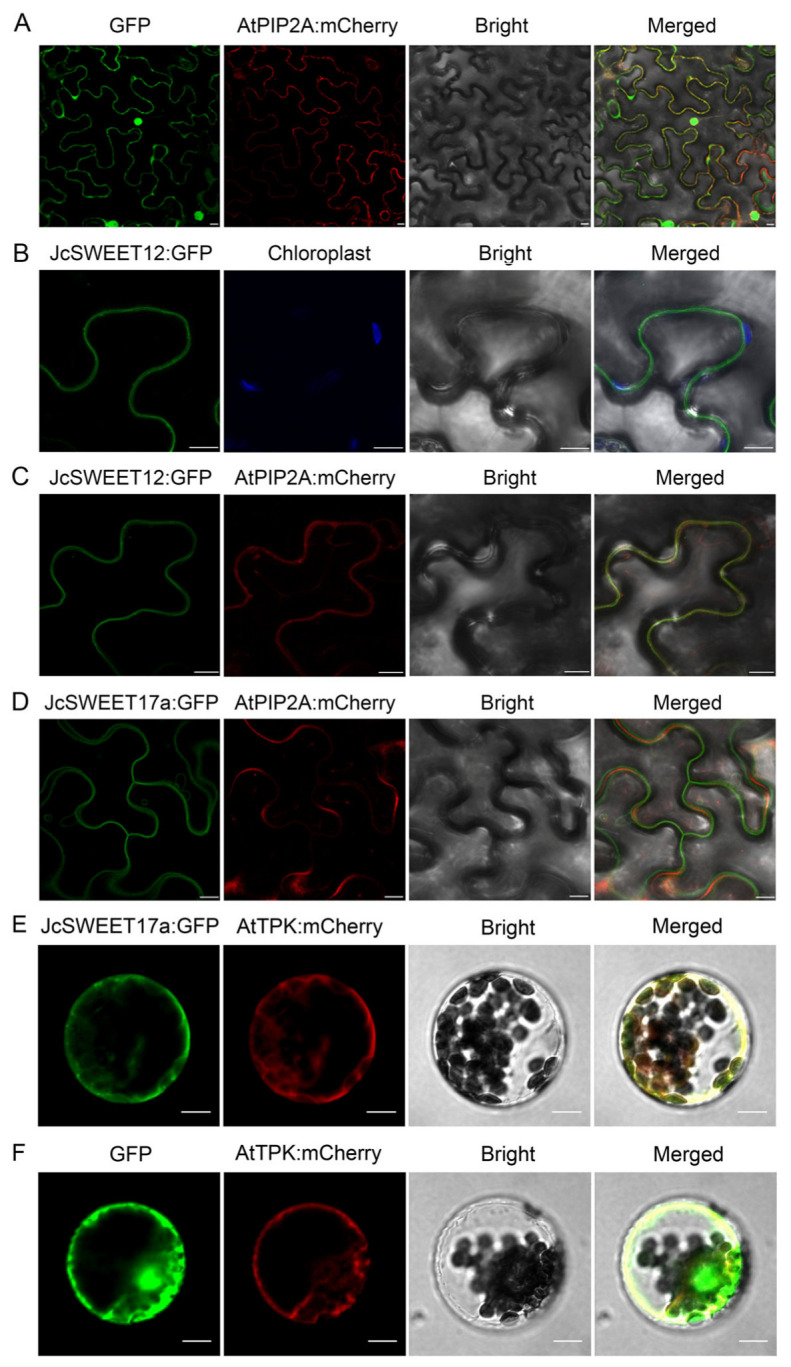
The subcellular localization of JcSWEET12 and JcSWEET17a in the epidermal cell of *N. benthamiana* leaves and Arabidopsis mesophyll protoplasts. (**A**) The co-expression of the positive control GFP protein and the plasma membrane marker protein of AtPIP2A. (**B**) The expression of the JcSWEET12:GFP fusion protein. The blue fluorescence indicates the autofluorescence of the chloroplasts. (**C**) The co-expression of the JcSWEET12:GFP fusion protein and the plasma membrane marker protein of AtPIP2A:mCherry. (**D**) The co-expression of the JcSWEET17a:GFP fusion protein and the plasma membrane marker protein of AtPIP2A:mCherry. (**E**) The co-expression of the JcSWEET17a:GFP fusion protein and the tonoplast marker protein of AtTPK:mCherry. (**F**) The co-expression of the positive control GFP protein and the tonoplast marker protein of AtPIP2A:mCherry. Scale bars = 10 µm.

**Figure 2 ijms-25-08183-f002:**
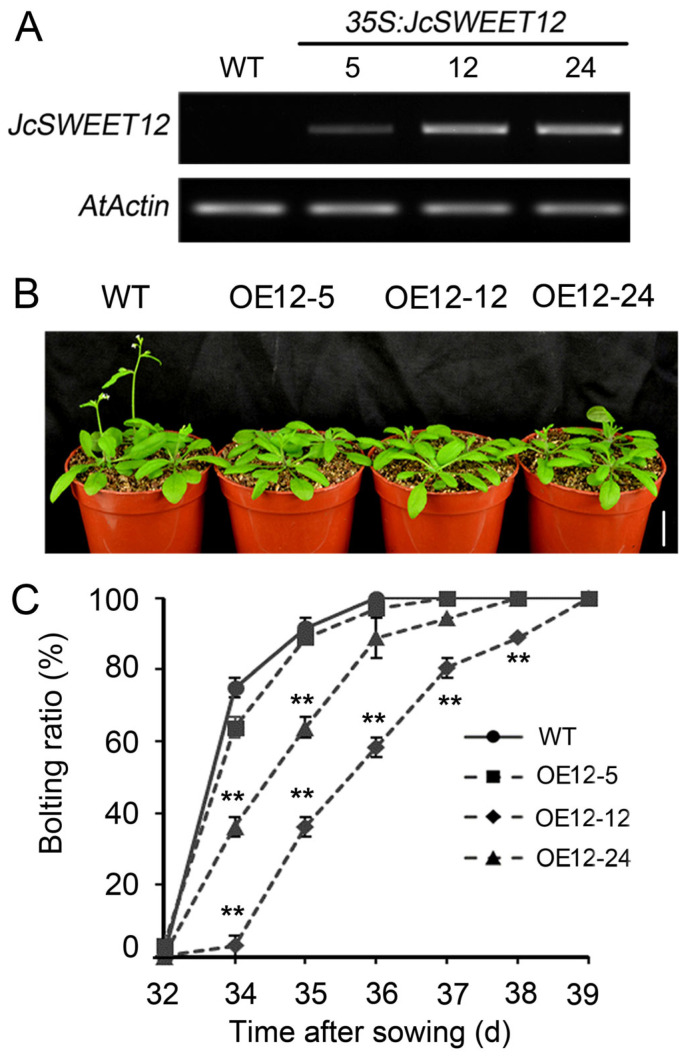
The late flowering phenotype of *JcSWEET12*-overexpressing Arabidopsis (OE12) plants. (**A**) The relative expression levels of *JcSWEET12* in different transgenic lines measured using semi-quantitative RT-PCR. (**B**) The initial bolting stage of WT and OE12 plants. Scale bar = 2 cm. (**C**) The bolting time of WT and OE12 lines. The data shown are means ± SD (*n* = 18). Statistically significant differences were assessed using Student’s *t*-tests (** *p* < 0.01).

**Figure 3 ijms-25-08183-f003:**
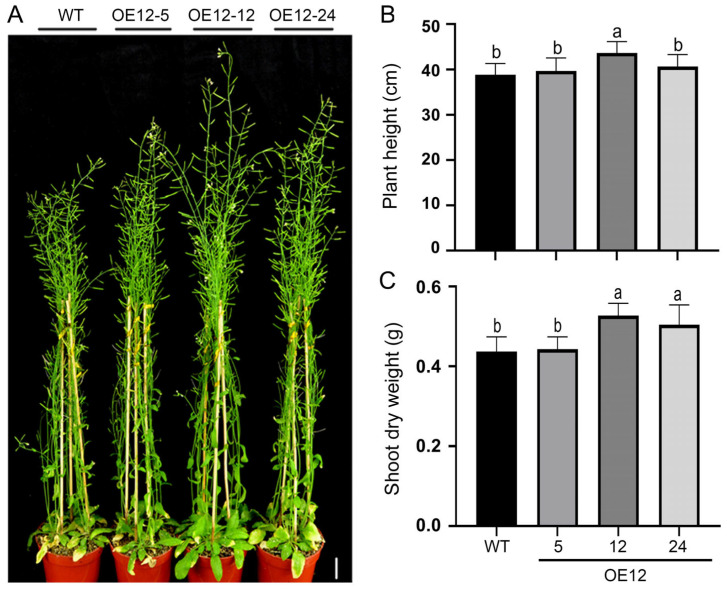
The characterization of plants of OE12 lines at the late development stage. (**A**) The phenotype of two-month-old plants. (**B**) Plant height at the two-month-old stage. (**C**) The dry weight of shoots at the two-month-old stage. The data shown are means ± SD (*n* = 18). Scale bar = 2 cm. Different letters above the bars denote significant differences according to one-way ANOVA with post hoc Duncan testing (*p* < 0.05).

**Figure 4 ijms-25-08183-f004:**
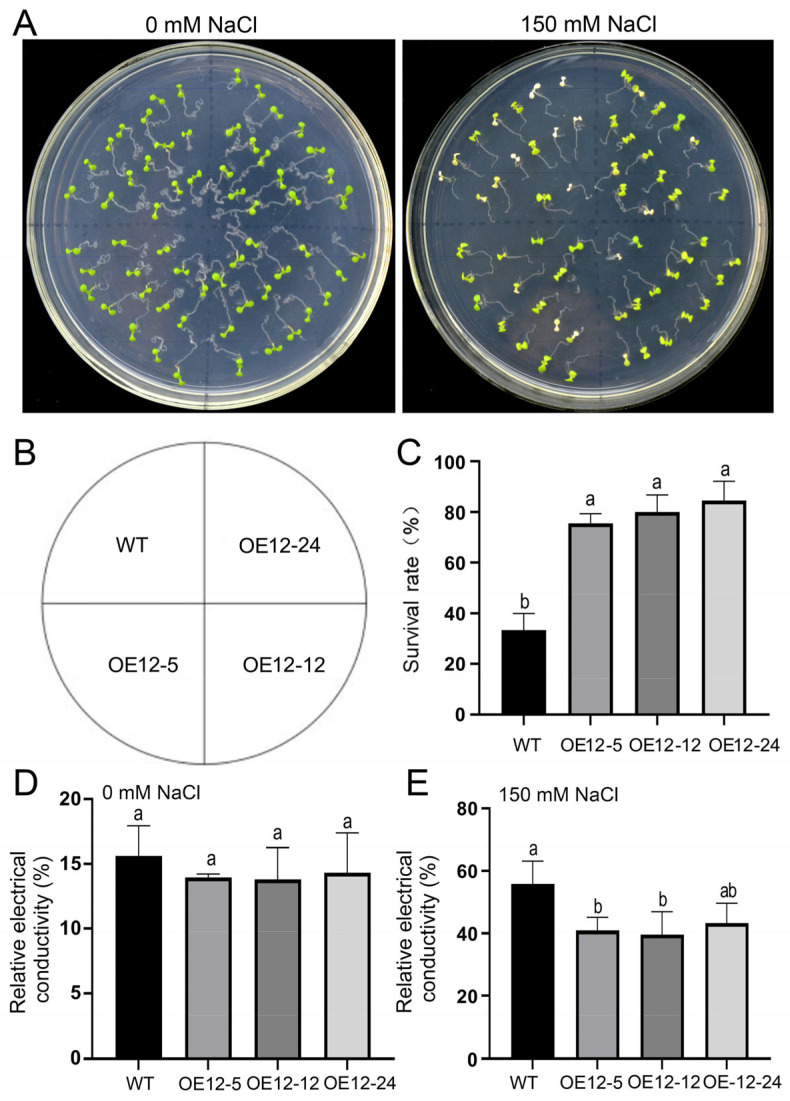
Overexpression of *JcSWEET12* (OE12) improves salt tolerance in Arabidopsis. (**A**) Seedlings after salt stress treatment. Four-day-old seedlings of WT and OE12 lines were transferred to 1/2 MS medium supplemented with 0 mM and 150 mM NaCl for 6 days. (**B**) Schematic representation of seedling position. (**C**) Survival rate of seedlings after salt stress. (**D**,**E**) Relative electrolyte leakage of WT and OE12 lines after salt stress treatment with 0 mM (**D**) and 150 mM NaCl (**E**). Data shown are means ± SD from three biological experiments (*n* = 15). Different letters above bars denote significant differences according to one-way ANOVA with post hoc Duncan testing (*p* < 0.05).

**Figure 5 ijms-25-08183-f005:**
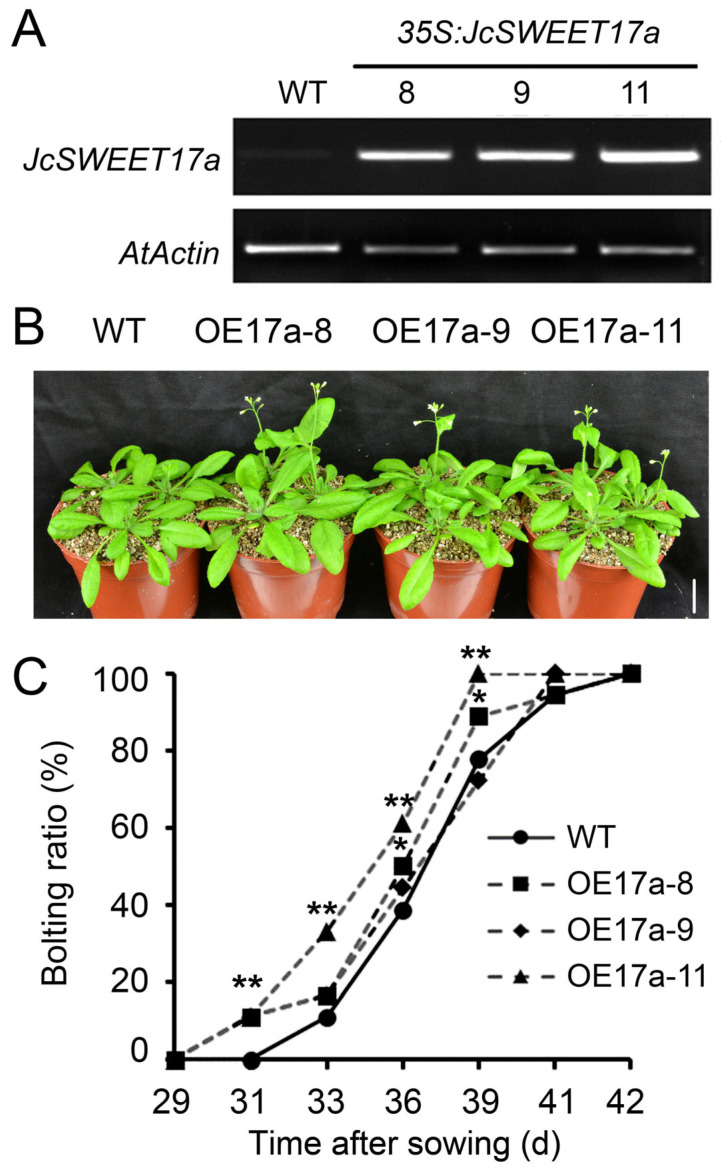
The flowering phenotype of *JcSWEET17a*-overexpressing (OE17a) Arabidopsis plants. (**A**) The relative expression levels of *JcSWEET17a* in different transgenic lines measured using semi-quantitative RT-PCR. (**B**) The initial bolting stage of plants of OE17a. Scale bar = 2 cm. (**C**) The bolting time of OE17a lines. The data shown are means ± SD (*n* = 18). Statistically significant differences were assessed using Student’s *t*-tests (* *p* < 0.05, ** *p* < 0.01).

**Figure 6 ijms-25-08183-f006:**
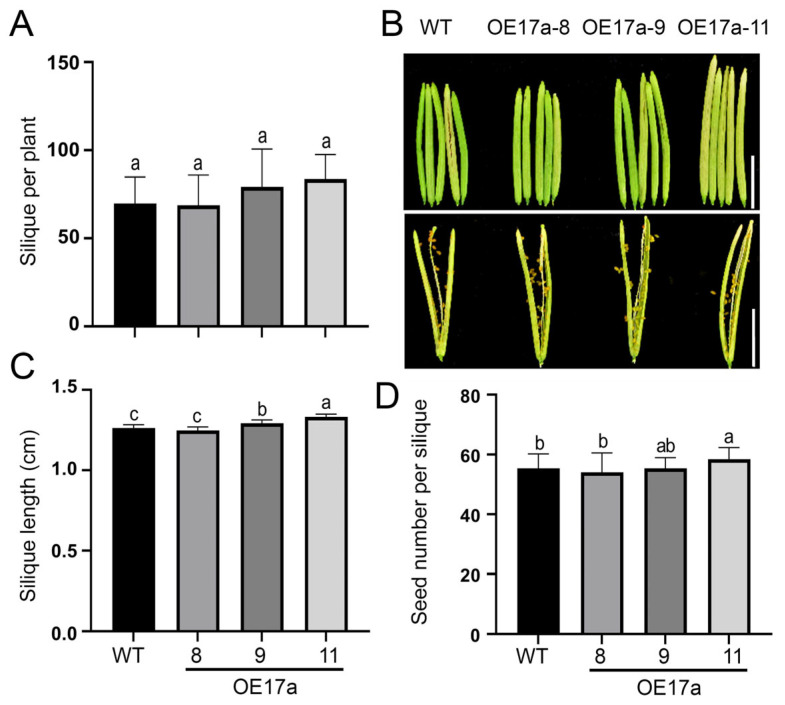
The silique characteristics of OE17a plants. (**A**) The average number of siliques per plant. The data shown are means ± SD (*n* = 18). (**B**) The size of siliques. Scale bar = 5 mm. (**C**) The average length of siliques. The data shown are means ± SD (*n* = 100). (**D**) The average seed number per silique. The data shown are means ± SD (*n* = 100). Different letters above the bars denote significant differences according to one-way ANOVA with post hoc Duncan testing (*p* < 0.05).

**Figure 7 ijms-25-08183-f007:**
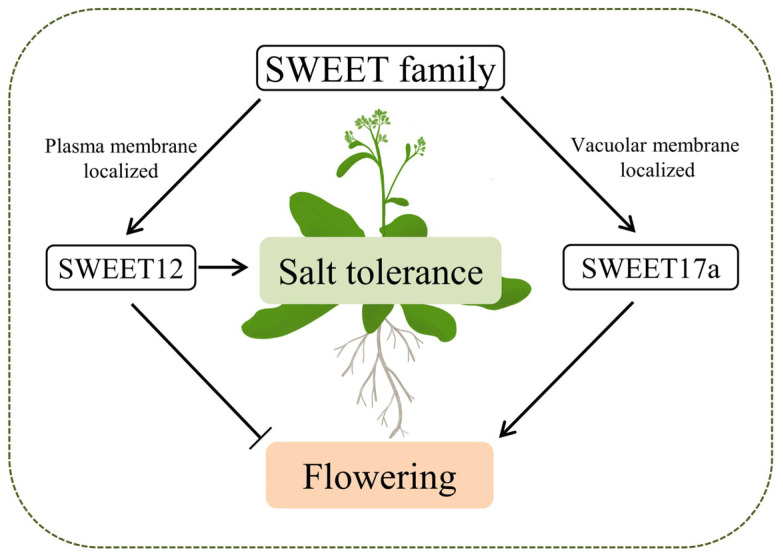
A model for SWEET-mediated regulation of plant development and abiotic stress tolerance.

## Data Availability

Data are contained within the article and Appendix A.

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
