# Peer review of "Functional Characterization of JcSWEET12 and JcSWEET17a from Physic Nut"

_ijms, 2024, doi:10.3390/ijms25158183_

Round 1
Reviewer 1 Report
Comments and Suggestions for Authors
In the paper, the author studied SWEET genes identified in their previous publication. However, the paper lacks sufficient data for publication in the International Journal of Medical Science. So, I would suggest major revisions before publication and suggest the following improvements:
Comment 1: Remove the plagiarism, as the iThenticate report indicates almost 36%.
Comment 2: In the abstract, provide a 1–1.5-line introduction about Jatropha curcas.
Comment 3: In the abstract, also provide 1-2 lines about the significance of sweet genes and why you have studied them.
Comment 4: Lines 55–56: Move these lines to the start of the introduction and combine them by writing more lines on the significance of Jatropha. At least a 5–6-line paragraph.
Comment 5: The study overall has very limited results at this stage. As it’s functional characterization, I would suggest adding gene expression profiles of both studied genes because, right now, it's very weird how we directly selected these two genes for study. So provide the expression profiles under salinity stress in jatropha plants.
Comment 6: Although the author reported about this family in his previous paper, I would suggest adding a phylogenetic tree of jatropha in comparison with model crops like Arabidopsis and tobacco. So the readers will know how many sweet family members are there in Jatropha. It will also add some value to the paper.
Comment 7: At this point, parameters related to stress and salinity are very limited in the paper, and only physiological parameters are reported. So, I would suggest adding various biochemical analyses of overexpression plants, including antioxidants like SOD, POD, and CAT, as well as some ROS contents in terms of H2O2.
Comment 8: The discussion part is very short and needs more details related to the parameters studied.
Comment 9: The authors must provide a detailed working model figure at the end of the paper to explain the role of both genes and the main results of the paper.
Author Response
Reviewer comments:
Responses to Reviewer1
Reviewer 1:
In the paper, the author studied SWEET genes identified in their previous publication. However, the paper lacks sufficient data for publication in the International Journal of Medical Science. So, I would suggest major revisions before publication and suggest the following improvements:
Comments 1: Remove the plagiarism, as the iThenticate report indicates almost 36%.
Response 1: Thanks for your comments and suggestion. We have revised the whole manuscript to reduce the repetition rate.
Comments 2: In the abstract, provide a 1–1.5-line introduction about Jatropha curcas.
Response 2: Thanks for your comments and suggestion. We have added “Physic nut (Jatropha curcas L.) has attracted extensive attention because of its fast growth, easy reproduction, tolerance to barren conditions and high oil content of seeds.” in the abstract.
Comments 3: In the abstract, also provide 1-2 lines about the significance of sweet genes and why you have studied them.
Response 3: Thanks for your comments and suggestion. We have added “contribute to regulate the distribution of carbohydrates in plants and have great potential in improving yield and stress tolerance.” in the abstract.
Comments 4: Lines 55–56: Move these lines to the start of the introduction and combine them by writing more lines on the significance of Jatropha. At least a 5–6-line paragraph.
Response 4: Thanks for your comments and suggestion. We have added a paragraph “As a kind of renewable alternative energy, biodiesel has attracted much attention because of its good environmental effect and great economic potential. The biomass enriched energy plant physic nut (Jatropha curcas L.), belonging to the Euphorbiaceae family, is a kind of plant suitable for the production of biodiesel, and is one of the most suitable energy plants for biodiesel feedstock [1,2]. It often appears as shrub or small perennial tree [2]. In addition, physic nut is a kind of multi-purpose, high light efficiency tree species, and suitable for planting in poor soil in marginal lands, these excellent characteristics make it widely concerned by people [3].” at the start of the introduction.
Comments 5: The study overall has very limited results at this stage. As it’s functional characterization, I would suggest adding gene expression profiles of both studied genes because, right now, it's very weird how we directly selected these two genes for study. So provide the expression profiles under salinity stress in jatropha plants.
Response 5: Thanks for your comments and suggestion.
Firstly, we chose these two genes for the following reasons: previous studies have shown that the SWEET gene regulates the distribution of carbohydrates in seeds and stress response, and has relatively significant potential value in creating varieties with enhanced storage or stress resistance. Therefore, in this manuscript, we selected a SWEET12 gene that is relatively highly expressed in seeds and SWEET17a gene that responds to drought/salt stress to carry out research, in order to obtain excellent gene resources. We have added “Many studies have shown that the SWEET gene regulates the distribution of carbo-hydrates in seeds and stress response, and has relatively significant potential value in creating varieties with enhanced storage or stress resistance.” and “In the current study, we selected a SWEET12 gene that is relatively highly expressed in seeds and SWEET17a gene that responds to drought/salt stress to carry out research, in order to obtain excellent gene resources.” at the last paragraph of introduction part.
Secondly, the expression patterns of these two genes in different tissues of physic nut (Figure 3), as well as the expression patterns of SWEET12 gene during seed development (Figure 4) and SWEET17a gene under abiotic stress (Figure 5) can be found at our previous paper (Wu et al. Genome-wide identification, expression patterns and sugar transport of the physic nut SWEET gene family and a functional analysis of JcSWEET16 in Arabidopsis. Int J Mol Sci. 2022, 23(10):5391).
Comments 6: Although the author reported about this family in his previous paper, I would suggest adding a phylogenetic tree of jatropha in comparison with model crops like Arabidopsis and tobacco. So the readers will know how many sweet family members are there in Jatropha. It will also add some value to the paper.
Response 6: Thanks for your comments and suggestion. Also in the previous paper, to investigate the phylogenetic relationships among the SWEET genes in physic nut and other plant species, a phylogenetic tree was constructed by aligning 18 JcSWEET protein sequences, 17 AtSWEET protein sequences from A. thaliana and 21 OsSWEET protein sequences from rice using the program MEGA5.0 (Figure 1). In order to test whether these tandem duplicates arose from recent duplication events in physic nut, we constructed another phylogenetic tree using SWEET proteins from physic nut, Arabidopsis and a closely related species, the castor bean (Ricinus communis) (Figure S4).
Comments 7: At this point, parameters related to stress and salinity are very limited in the paper, and only physiological parameters are reported. So, I would suggest adding various biochemical analyses of overexpression plants, including antioxidants like SOD, POD, and CAT, as well as some ROS contents in terms of H2O2.
Response 7: Thanks for your comments and suggestion. In this manuscript, we mainly found that these two genes (SWEET12 and SWEET17a) have opposite effects on flowering regulation. Surprisingly, SWEET12 gene can enhance the tolerance to salt stress, while SWEET17a does not. The mechanism of their regulation is what we will focus on next. Furtherly, we will study whether their functions are closely related to subcellular localization/transport substrates and other factors, and to find out the changes of specific biochemical indicators.
Comments 8: The discussion part is very short and needs more details related to the parameters studied.
Response 8: Thanks for your comments and suggestion. In our revised manuscript, we enrich the discussion section.
Comments 9: The authors must provide a detailed working model figure at the end of the paper to explain the role of both genes and the main results of the paper.
Response 9: Thanks for your comments and suggestion. We have provided a detailed working model figure at the end of the paper to explain the role of both genes and the main results of our manuscript (Figure 7).
Reviewer 2 Report
Comments and Suggestions for Authors
Functional characterization of JcSWEET12 and JcSWEET17a 2 from physic nut
This is an interesting manuscript, however, several questions, and suggestions could improve its content.
First, details of the experimental designs must be included like the experimental design layout, experimental units, and treatments. Paragraphs from 2.2 to 2.5, include the bolting ratio through time (Figures 2C and 5C) where you have an experiment of repeated measurements. The statistical analysis must consist of the ANOVA where the factors, treatments, time, and interaction must be included. In my opinion, a nonlinear regression of a sigmoid curve of each treatment must also be done as the growth rate's derivation.
For the other variables, plant height, shoot weight, silique per plant, silique length, survival rate, and seed number per silique to compare the treatments, besides the ANOVA, a pairwise comparison must be made.
Relative electrical conductivity must be a factorial experiment including the treatments and amount of mM NaCl and interaction; pairwise comparisons must be provided.
Author Response
Reviewer comments:
Responses to Reviewer2
Reviewer2:
This is an interesting manuscript, however, several questions, and suggestions could improve its content.
Comments 1: First, details of the experimental designs must be included like the experimental design layout, experimental units, and treatments. Paragraphs from 2.2 to 2.5, include the bolting ratio through time (Figures 2C and 5C) where you have an experiment of repeated measurements. The statistical analysis must consist of the ANOVA where the factors, treatments, time, and interaction must be included. In my opinion, a nonlinear regression of a sigmoid curve of each treatment must also be done as the growth rate's derivation.
Response 1: Thanks for your comments and suggestion. According to the suggestion, we re-analyzed the data and modified the figures and corresponding text content. The figure of flowering time is too messy to be presented with ANOVA, so the T-test is still used, and I hope reviewers can understand my explanation.
Comments 2: For the other variables, plant height, shoot weight, silique per plant, silique length, survival rate, and seed number per silique to compare the treatments, besides the ANOVA, a pairwise comparison must be made.
Response 2: Thanks for your comments and suggestion. According to the suggestion, we re-analyzed the data and modified the figures and corresponding text content.
Comments 3: Relative electrical conductivity must be a factorial experiment including the treatments and amount of mM NaCl and interaction; pairwise comparisons must be provided.
Response 3: Thanks for your comments and suggestion. According to the suggestion, we re-analyzed the data and modified the figures and corresponding text content.
Round 2
Reviewer 1 Report
Comments and Suggestions for Authors
I would like to appreciate authors for making appropriate revisions. The manuscript is revised according to my suggestions, so I will suggest acceptance of the paper in its present form.
Reviewer 2 Report
Comments and Suggestions for Authors
It would be more informative to adjust sigmoidal curves for the Bolting ratio.